# Radiosurgery for Intracranial Meningiomas: A Review of Anatomical Challenges and an Update on the Evidence

**DOI:** 10.3390/cancers17010045

**Published:** 2024-12-27

**Authors:** Matthew J. Goldman, Bin S. Teh, Simon S. Lo, E. Brian Butler, David S. Baskin

**Affiliations:** 1Department of Neurological Surgery, Houston Methodist Neurological Institute, Houston Methodist Hospital, Houston, TX 77030, USA; mjgoldman@houstonmethodist.org; 2Department of Radiation Oncology, Houston Methodist Hospital, Houston, TX 77030, USA; bteh@houstonmethodist.org (B.S.T.); ebutler@houstonmethodist.org (E.B.B.); 3Departments of Radiation Oncology and Neurological Surgery, University of Washington, Seattle, WA 98195, USA; simonslo@uw.edu; 4Kenneth R. Peak Brain and Pituitary Tumor Treatment Center, Houston Methodist Hospital, Houston, TX 77030, USA; 5Department of Neurological Surgery, Weill Cornell Medical College, New York, NY 10022, USA; 6Department of Medicine, Texas A & M Medical School, Houston, TX 77030, USA

**Keywords:** radiation, meningioma, brain tumor, neurosurgery, radiation oncology, anatomy

## Abstract

Meningiomas are slow growing and often benign brain tumors. Radiation is a non-invasive alternative to treat meningiomas, traditionally reserved for poor surgical candidates, Grade II or III tumors, or those with residual or recurrent lesions following surgery. In the past three decades, stereotactic radiosurgery (SRS) has emerged as a standard treatment for meningiomas, especially for small or incidentally discovered tumors. Over the past decade, increasingly specific anatomic-based publications have described nuances and the efficacy of radiosurgical intervention at specific anatomical subsites. This narrative review describes subsite anatomy, new evidence, and future research directions.

## 1. Introduction

Meningiomas have been well described over centuries but received their formal name by the seminal work of Harvey Cushing in 1922 [1]. Defined as extra-axial masses originating from arachnoidal cap cells of the leptomeninges [2], these tumors grow within the brain’s outer covering of the arachnoid mater and pia mater.

Meningiomas are the most common primary central nervous system (CNS) tumors, accounting for 15% of symptomatic intracranial neoplasms and 33% of asymptomatic intracranial neoplasms found on autopsy [3,4]. They disproportionately impact women, with 65% found in women aged 40–50 years [4].

Definitive diagnosis requires biopsy or resection; however, typical radiologic appearance, the most common diagnostic technique, often suffices [1]. Magnetic resonance imaging (MRI) findings classically show a dural-based, well-circumscribed, homogeneously enhancing lesion [1]. Radiographic classification distinguishes en plaque meningioma (extensive dural thickening) from globose meningiomas (rounded extra-axial masses) [4].

The World Health Organization (WHO) acknowledges 15 histological subtypes of meningiomas, predominantly WHO Grade I (benign). Despite their benign nature, encapsulated structure, and limited genetic aberrations, their anatomical locations may lead to serious and potentially lethal consequences [5]. WHO Grade II and III meningiomas have higher recurrence rates and require a more aggressive treatment. Certain molecular factors indicate malignant behavior, including telomerase reverse transcriptase (TERT) promoter mutation or homozygous deletions in cell cycle regulator genes CDKN2A and/or CDKN2B [6].

Meningiomas are typically slow-growing and non-infiltrative lesions with an insidious onset of site-dependent symptoms. While lacking pathognomonic presentation, these neoplasms may manifest with common intracranial tumor symptoms, including headache from increased intracranial pressure, focal neurological deficits, and seizures [1]. Brain edema, mass effect, or direct compression of critical structures drive meningioma symptoms. Despite their high prevalence, clinical management varies significantly based on multifaceted considerations, including location, size, patient age, presentation, and pathology [7].

Treatment options for intracranial meningiomas encompass surgical resection alone, surgery followed by adjuvant radiation therapy (RT), or primary RT [8]. The treatment of choice, when safely attainable, is complete surgical resection (Simpson Grade I–III), including the dura of origin and infiltrated bone [9]. However, inherent surgical risk may preclude complete resection, particularly for tumors infiltrating vital intracranial structures or within frail patient populations. Subtotal resection and higher grade (WHO II and III) have higher rates of recurrence; multiple resections increase morbidity secondary to scarring and increased infiltration [10]. Since the 1980s dawn of radiotherapy planning via magnetic resonance imaging (MRI) or computerized tomography (CT), the 5-year progression-free survival rate in benign meningiomas increased from 77% to 98% [11]. When complete surgical resection was unfeasible, subtotal resection with adjuvant radiation can achieve results comparable to total resection [11].

Radiotherapy (RT) can be delivered to intracranial meningiomas in different modalities, including stereotactic radiosurgery (SRS), hypofractionated stereotactic radiotherapy (HSRT), and conventional radiotherapy. SRS utilizes a single irradiation fraction, while HSRT delivers 2–5 ablative doses of focused radiation. Conventional radiotherapy entails daily delivery of small individual radiation doses over 5–6 weeks.

Historically, both SRS and FSRT were reserved for recurrent or malignant lesions, or in patients who were poor candidates for surgical intervention [9,12]. Increased frequency of brain imaging, together with an aging population, has permitted the diagnosis of asymptomatic meningiomas [13,14]. RT has become a first-line therapy for select meningioma cases, particularly small meningiomas in challenging or high-risk anatomical locations [15]. Combination treatments with surgery and SRS/FSRT are also increasingly used, but controversies remain regarding timing, type, and dosing therapy of various RT techniques [16].

Population-based analysis reveals increased incidence over recent years alongside decreasing tumor size at diagnosis time [7]. A metanalysis on meningioma growth showed that although half of incidental meningiomas exhibit no growth, the remaining half exhibited either linear or exponential growth [17]. Strategic surgical decision-making hinges on precise meningioma origin determination and anatomical considerations [18]. Adjacent anatomical structures dictate ease of surgical access, ability to achieve complete resection, and risk of morbid tumor progression.

Over recent years, significant research focusing on RT for meningioma control has been shared among the literature. While prior investigations detailed outcomes in broader regions, such as the skull base and parasellar regions, recent reports have accumulated detailing outcomes, nuances, and updated SRS treatment regimens for specific anatomical subregions; instead of a study investigating skull-based meningiomas broadly, more specific analyses now investigate subregions, such as petroclival [19,20,21], olfactory groove [22,23,24], and foramen magnum [25,26,27,28].

This narrative review will discuss anatomical challenges inherent in specific locations and synthesize the latest evidence on radiosurgical intervention. We will then delve into intricacies of radiosurgical methodology, dosimetric considerations, and future directions of radiotherapy.

## 2. Methodology

Using a narrative review framework, we searched the recent literature (2000–2024) using the MEDLINE database with search keywords “meningioma”, “radiosurgery”, and each anatomic location individually. We also included relevant articles fitting this time period (2000–2024) noted in ascendency searching. Articles were included if they described radiosurgery for meningiomas, especially if they described anatomical location. Articles were excluded if they were not translated into English, available in full text, or discussed meningioma in broad anatomic descriptions.

## 3. Classification by Location

Meningiomas are named based on their anatomical location. The most common sites reported are falcine, parasagittal, convexity, and sphenoid wing meningiomas [4,29]. Their presentation entirely depends on adjacent neuronal and vascular structures impacted by mass effect. Prior reviews have reported the most common presenting symptoms: headache (33.3–36.7%), focal cranial nerve deficit (28.8–31.3%), seizure (16.9–24.6%), cognitive change (14.4%), weakness (11.1%), vertigo/dizziness (9.8%), ataxia/gait change (6.3%), pain/sensory change (5.6%), proptosis (2.1%), and syncope (1.0%). A significant minority are asymptomatic (9.4%) [30].

Although broad symptoms like headache are shared, Table 1 details specific location-based presenting symptoms. With modern imaging advances, meningiomas are being discovered earlier and smaller than ever before [7]. Early-stage diagnoses often lack the locational symptoms seen with larger presenting tumors.

As further research emerges on specific anatomical locations, results can be interpreted on a more specific basis than prior broad encompassing regions. For example, prior results have been posted on radiosurgery for “skull base” meningiomas, but this encompasses subregions like the clivus, petrous bone, cavernous sinus, tentorium, sphenoid bone, sella turcica, olfactory groove, and optic nerve sheath [9]. This paper reviews evidence for radiosurgery within these specific anatomical subregions, highlights regional challenges, and provides a depth of available outcome-based evidence on tumor control. Figure 1 highlights regions covered within this review and their relation to one another.

## 4. Guidelines for Meningioma Treatment

Meningioma management lacks a clear-cut algorithm; an appropriate management strategy is determined on a case-by-case basis [6]. Despite being a common intracranial tumor, countries and surgical centers vary in management approaches [16]. Trends derived from the Surveillance, Epidemiology, and End Results (SEERs) database of the National Cancer Institute in 2017 showed that 42.4% of patients with meningiomas underwent surgical treatment with or without radiotherapy, while 52.1% were managed with active surveillance [7]. Of those managed surgically, 8.7% received RT at some point, with 95% receiving it after surgery. Only 5.6% of patients underwent radiation therapy alone [7].

While some meningiomas may not increase in size, many do if watched long enough, especially in young patient populations. It is statistically unlikely that a 20-year-old will have 50 growth-free years; therefore, treatment decision-making must consider age [37]. Prior publications on the natural history of meningioma growth have shown exponential or linear growth in 63% [38] and 53% [39], and overall growth in up to 83.5% of tumors [40]. Considering growth potential is particularly relevant for small asymptomatic meningiomas adjacent to vasculature or cranial nerves.

Recently, increased investigation has examined RT as primary monotherapy instead of conservative observation. An important study by Sheehan et al. compared SRS alone vs. active surveillance in asymptomatic incidental meningiomas. Their analysis found SRS affords superior radiologic tumor control versus active surveillance without an increasing risk of neurological deficits [41].

Updated 2021 European Association of Neuro-Oncology (EANO) guidelines list observation as the first option for asymptomatic meningiomas. These guidelines also include RT as complementary therapy or even an alternative approach to surgery in certain situations [16].

## 5. Anatomical Locations and Updated Evidence

### 5.1. Parafalcine and Parasagittal

The falx cerebri is a sickle-shaped structure running within the longitudinal fissure between the two cerebral hemispheres. Anchored anterior to the crista galli and posterior to the occipital protuberance, it forms a sail-like divider of the two hemispheres [42]. Meningiomas originating from the falx are termed parafalcine meningiomas [43]. If the meningioma grows into at least one wall of the superior sagittal sinus, it is referred to as a parasagittal meningioma [44].

Parafalcine and parasagittal meningiomas account for 30% of intracranial meningiomas [45]. Microsurgery is the mainstay of meningiomas at this location, despite high recurrence and the risk of venous injury during surgical excision [45]. Engulfment of the pericallosal arteries deters radical excision, of which radiotherapy could be useful [31]. With high rates of venous invasion, complete resection is generally unfeasible [45]. Optimal management remains controversial because both surgical and non-surgical options frequently fail to provide definitive treatment [10].

A review by Pinzi et al. in 2019 found limited literature on SRT and RS for parafalcine and parasagittal meningiomas [45], with only five available studies and only one study since 2015 [41]. Their analysis found falcine tumors were suitable for complete resection, while parasagittal meningiomas were difficult due to sagittal sinus adherence [45].

It has been reported previously that parasagittal and parafalcine meningiomas have suboptimal radiosurgical response versus skull-based lesions [41]. Pinzi et al. concluded from their data review that SRT and RS could be more widely applied for parafalcine and parasagittal meningiomas [45], representing viable options for primary and adjuvant therapy for parasagittal and parafalcine meningiomas [45].

Ding et al. published a retrospective analysis of 65 patients treated with a median prescription dose of 15 Gy for parasagittal and parafalcine meningiomas in one fraction. They found a tumor control rate of 85% and 70% at 3 and 5 years. When excluding treated tumors (resection, embolization, and radiotherapy), they found the control rate at both 3 and 5 years was 93% [10]. This suggests that prior treatment negatively affected radiosurgical efficacy, but additional studies are needed. Parasagittal location, no prior resection, and younger age were independent predictors of tumor progression-free survival [10].

Meningiomas at the confluence of the falx and tentorium (CFT) represent <2% of intracranial meningiomas and are surgically challenging [32]. Abdallah et al. published a paper reporting the largest series focused exclusively on outcomes of Gamma Knife radiosurgery for meningiomas arising from the confluence of the CFT in 2022 [32]. Their paper detailed 20 CFT meningioma patients, 10 patients undergoing radiosurgery for progression of residual tumor following resection and 10 undergoing radiosurgery as primary management. The marginal and maximum doses delivered in this series were 13.0 Gy (IQR 12.1–14.8) and 26 Gy (IQR 25.0–29.5), respectively. All radiation was carried out with a single unfractionated session. The 1, 5, and 10-year local tumor control rates were 100% (N = 20), 100% (N = 10), and 83% (N = 4), respectively [32]. Tumor volume decreased in 11 patients and was stable in eight patients. Their study concluded that radiosurgery is a safe and effective treatment for CFT meningiomas that can be utilized not only as adjuvant therapy, but also for primary management [32].

Together, these studies suggest that radiosurgical intervention for parasagittal meningiomas may be more effective than parafalcine. There was good tumor control overall and acceptable rates of adverse events. More studies are warranted on this subgroup of meningiomas to determine optimal treatment.

### 5.2. Convexity

Convexity meningiomas are predominate among extra-axial tumors encountered in neurosurgery [46]. They grow on the outer surface of the cerebrum directly adjacent to the flat bones of the skull (see Figure 1). These tumors are relatively simple to approach and resect because both the mass and involved dura mater can be removed [47], but account for only about one-sixth of meningiomas [1]. Craniotomy and removal of convexity meningiomas along with their dural base is the treatment of choice for symptomatic patients [47]. Despite easier surgical access, the optimal treatment for asymptomatic convexity meningiomas remains unclear, with options including active surveillance or SRS. Observation is the most commonly preferred initial treatment; however, 4.6–26.3% of these asymptomatic patients will develop symptoms and require treatment [48]. The role of SRS for upfront treatment of asymptomatic convexity meningiomas has yet to be well defined [48].

Pikis et al. in 2022 investigated data from patients with asymptomatic convexity meningiomas at 14 participating centers using a focused analysis from the International Multicenter Matched Cohort Analysis of Incidental Meningioma Progression During Active Surveillance (IMPASSE) study [48]. Their study examined two unmatched cohorts of patients with asymptomatic convexity meningiomas: one received upfront SRS (N = 99), and one was managed conservatively (N = 140). The mean marginal dose of radiation was 13.26 Gy. After propensity matching for age, they compared 98 patients in each cohort. A matched analysis showed that tumor control was 99% in SRS-treated patients and 69.4% in patients managed conservatively. Their results suggested that the upfront SRS treatment of asymptomatic convexity meningiomas is associated with increased tumor control without an increased risk of new neurological deficits.

Another group of patients with convexity meningiomas underwent a retrospective review by Ruiz-Garcia et al. in 2021. They analyzed 18 patients with neurofibromatosis type 2 (NF2) with convexity meningiomas. One hundred and twenty convexity meningiomas were treated with single-fraction radiosurgical treatment, with a median marginal dose of 12.0 Gy. Actuarial tumor control rates at 5, 10, and 15 years were 100% with only one radiation-induced adverse event [49].

Overall, the studies reviewed suggest that upfront SRS may offer superior control over watching/waiting. Convexity meningiomas lack extensive evidence on SRS. Symptomatic lesions are highly accessible for surgical intervention, but small asymptomatic incidental convexity meningiomas may benefit from SRS over watchful waiting. Additionally, patients with a high tumor lifetime burden, similar to those with NF2, may benefit from limiting invasive intervention to critical only. Fewer craniotomies may preserve quality-of-life in NF2 patients [49].

### 5.3. Cavernous Sinus

The cavernous sinus is a bilateral structure located lateral to the sella turcica and extends from the superior orbital fissure anteriorly to the petrous part of the temporal bone posteriorly [50]. Complete surgical resection of cavernous sinus (CS) meningiomas is quite difficult due to the numerous important regional structures and thus carries significant patient risk [51]. The cavernous sinus houses a segment of the internal carotid artery (ICA), the sympathetic plexus, cranial nerves III, IV, VI, and the first two branches of CN V. Additionally, a narrow surgical corridor and close relation with the optic chiasm further complicate open resection [52]. Microsurgical operations may cause cranial nerve palsies, ICA injury or occlusion, and/or cerebrospinal fluid leak [53].

Anatomical localization of vital structures, lack of arachnoid surfaces, and the tendency of CS meningiomas to invade nerve fascicles make surgical cure extraordinarily difficult to achieve [52]. Achieving complete resection depends on the degree of ICA involvement and vascular wall invasion [52]. The available literature shows a tumor growth control rate of CS meningiomas >90%, but ongoing debate still exists regarding the optimal treatment; no class 1 evidence exists supporting the superiority of SRS alone, fractionated radiotherapy alone, or any combination of either with subtotal resection [54].

Basak et al. sought to compare the efficacy of surgery and Gamma Knife therapy with Gamma Knife therapy alone for cavernous sinus meningiomas in 2023. This study uniquely compared two treatment methodologies at a single institution with nine having surgery and radiosurgery, and eleven with radiosurgery alone; a mean marginal dose of >13 Gy in a single fraction was used in both groups. Extraocular nerve palsy is the most common complication with tumor management, a disturbing detriment to patient quality-of-life [52]. Their results showed this complication was more likely with surgical excision; recovery was more likely with GK treatment alone. Cranial nerves within the cavernous sinus seem to tolerate relatively higher doses than the optic apparatus [9], but optic nerve complications should be considered due to its radiation sensitivity. Keeping radiation to the optic nerve lower than 8–10 Gy may reduce the risk of adverse radiation effects [55].

After Basak et al. reviewed the available literature and their own cohort, they concluded that a distance of >3 mm between the tumor and optical anatomical structures is required for safe and effective radiosurgical intervention. They added that tumors with a volume of >10 mL and less than 3 mm margin with the optical apparatus would benefit from surgical volume reduction prior to SRS [52].

A larger meta-analysis was performed by Sughrue et al. including 2065 patients with CS meningiomas; 435 were treated with surgery alone, 71 with surgical subtotal resection followed by SRS, 1300 with SRS alone, and 250 undergoing fractionated radiotherapy alone. Details on radiation parameters were not provided within the publication. The authors found better recurrence rates among SRS patients (3.2% [95% CI 1.9–4.5%]) versus gross-total resection (11.8% [95% CI 7.4–16.1%]) or subtotal resection alone (11.1% [95% CI 6.6–15.7%]) (*p* < 0.01) [54]. Additionally, cranial neuropathy rates were markedly higher in patients undergoing resection (59.6% [95% CI 50.3–67.5%]) than SRS alone (25.7% [95% CI 11.5–38.9%]) (*p* < 0.05) [56].

Zeiler et al. performed an institutional retrospective review on CS meningiomas from 2003 to 2011. The maximum dosage to the cavernous sinus used was 30–40 Gy and <8 Gy for optic apparatus dosing. The average dose at the 50% isodose line was 13.5 Gy. Their study included 30 patients: 12 patients had previous surgical debulking, with only one patient asymptomatic at intervention time. The average follow-up was 36.1 months, with 26 total patients. Tumor size was reduced in 34.6%, stable in 57.7%, and increased in 7.7% [53]. Most complications were minor and transient.

A larger institutional retrospective study was carried out on 290 CS meningioma patients from 1987 to 2009 by Park et al. Gamma Knife radiosurgery was performed with a median marginal dose of 13 Gy (range 10–20 Gy) in a single session. The maximum optic pathway dosage was ≤10 Gy. Their review had an especially long follow-up period providing insight on long-term control. Follow-up was carried out at 1, 5, 10, and 15-year appointments, and tumor control rates were 98%, 93%, 85%, and 85%, respectively. In addition to excellent long-term control, they found that patients who underwent SRS without prior surgery or soon after microsurgery had a lower risk of treatment failure than patients undergoing SRS for progressive tumors [51].

Another institutional retrospective review was carried out by Spiegelmann et al. using a Linear Accelerator to treat 102 patients from 1993 to 2007. The minimal marginal single-session dose was 12–17.5 Gy and optic apparatus <10 Gy. The mean follow-up was 67 months and showed tumor control of 98% with 58% reducing volumes [57]. Their data strongly suggested early radiosurgery (<1 year from onset) without surgical intervention increases the chance of improving CN deficits. Close to 60% of deficits improved or resolved. The authors regarded radiosurgery as the treatment of choice for CS meningiomas [57]. These results corresponded with those from Kimball et al. where local tumor control was 100% at 5 years and 98% at 10 years [58]. The authors found RS offered greatly superior tumor control with lower morbidity than surgical resection [58].

The overall evidence above suggests radiosurgery is an effective treatment for halting cavernous sinus meningioma progression with acceptable complication rates. Evidence suggests SRS may decrease the incidence of cranial nerve palsies when compared with open surgical resection [52,56,57] and that there are potential benefits to upfront SRS without prior surgical intervention [51].

### 5.4. Parasellar

Parasellar meningiomas represent 15% of all meningiomas [18]. These meningiomas encompass a subset of skull base meningiomas in or around the sella turcica. This anatomically complex region forms the junction of important neurovascular structures [33]. Complete resection of these tumors can jeopardize adjacent vital structures, including the internal carotid artery, and optic, oculomotor, and trigeminal nerves [18,59]. Considering surgical risk, the goals of treatment value local tumor control over oncological cure. A systematic review of parasellar meningiomas showed new or progressive symptoms 61% of the time [60], suggesting some form of intervention is warranted. While the parasellar region includes the cavernous sinus, Meckel’s cave, petroclival region, optic canal, and the anterior clinoid [18], these specific parasellar subregions will be discussed individually in subsequent sections.

Matoušek et al. investigated a combination of endoscopic transnasal optic nerve decompression (ETOND) followed by SRS for parasellar meningiomas [33]. Their study included 12 patients. Nine of the patients underwent fractionated radiosurgery, with most receiving five fractions totaling 30 Gy of radiation. Visual acuity was noted in 10 of 14 eyes (71.4%) in 8 of 12 patients (66.7%). ETOND prior to SRS appeared to improve rates of visual function and reduce rates of SRS complications.

Hu et al. sought to investigate if prior surgery affected treatment outcomes of patients with parasellar meningiomas. Their study investigated 93 patients, 45 with surgery prior to Gamma Knife surgery. The median marginal dose was 12 Gy for both groups, in a single session. Patients were more likely to have improvement of preexisting symptoms without prior surgery (*p* = 0.009) and more likely to have stable symptoms with prior surgery (*p* = 0.012) [61].

Cohen-Inbar et al. investigated the influence of SRS treatment parameters and timing on parasellar meningiomas with long-term volumetric follow-up evaluation. One hundred and eight nine patients received a mean marginal single-session dose of 14 Gy (12–70 Gy), achieving tumor control in 91.5% of patients [59]. Results showed a significant difference in control rates between patients receiving <16 Gy compared to >16 Gy. At 15-year follow-up, the lesser group had a progression-free survival rate of 79.4%, compared to 95.7% in those that received >16 Gy [59]. Tumor volume at initial SRS being >14 cm^3^ predicted failed tumor control and early follow-up volumetric analysis of size change predicted long-term volume changes.

Williams et al. performed a retrospective review of 138 patients with parasellar meningiomas treated from 1989 to 2006. Notably, 84 out of 138 patients had undergone previous resection. The mean marginal dose used was 13.7 Gy (range 4.8–30 Gy). There was a low incidence of neurological defects and acceptable rates of tumor control within this cohort. Progression-free survival at 5 and 10 years was 95.4% and 69%, respectively [62]. Multivariate analysis showed that younger age at the time of intervention was a predictor of effective tumor control.

Sheehan et al. conducted a 10-center study from 1988 to 2011, with retrospective analysis of 763 patients with sellar and parasellar meningiomas treated with Gamma Knife radiosurgery [41]. Patients received a mean marginal dose of 13.2 Gy in a single session. Results showed a 90.2% overall tumor control rate and tumor progression-free survival rates at 3, 5, 8, and 10 years were 98%, 95%, 88%, and 82%, respectively. Contrary to other studies showing prior surgical resection predicted progression, this study found prior surgery favorable for tumor control. Smaller tumors and higher marginal dose favored tumor control.

For parasellar meningiomas, a distance of 5 mm between the meningioma and the optic nerve may be considered safe in single-shot RS [55]. Most studies limited radiation to <8 Gy. Taken together, these parasellar meningioma studies highlight radiosurgery as an effective form of intervention with low complication rates.

There was mixed evidence on outcomes for previous surgical interventions in SRS. One study suggested more likely symptomatic improvement from SRS without prior surgery [61]; another study found prior surgery favored tumor control [60]. Intervening in younger patients favored tumor control [62].

More evidence from larger patient populations is needed to determine prior surgery and other factors on SRS outcomes for parasellar meningiomas. Overall, SRS for parasellar meningiomas proved to be an effective upfront and adjuvant treatment with adequate rates of tumor control and minimal neurological complications.

### 5.5. Perioptic

Perioptic meningiomas are those less than 3 mm from the optic apparatus [63]. Serial imaging suffices for small asymptomatic lesions, although even minor growth can lead to visual deterioration or complete blindness [63]. Increasingly, practitioners are supporting radiotherapy for optic nerve sheath meningiomas for better visual outcomes [64]. Still, heterogenous literature evidence constrains surgical decision-making [64]. For many tumors within the optic nerve sheath, surgery has unacceptable outcomes, as tumor and optic nerve may share the same blood supply [65]. This makes SRS a good choice. Radiation must balance between stopping tumor progression and preventing radiation-induced optic neuropathy (RION); both may lead to compromised vision. Specific locations of tumors—like clinoid process, cavernous sinus, and parasellar region—may compress the optic apparatus, but these will be discussed in independent sections.

A systematic review carried out in 2018 by Hénaux et al. reported visual outcomes for surgical management of meningiomas compressing the optic nerve [64]. They reviewed 317 patient cases of meningiomas compressing the optic nerve from 2004 to 2012. Sixty-one patients had residual tumor, and 23 received postoperative radiotherapy. Researchers investigated how location-specific compression along the optic nerve course should influence operative decision-making. Their analysis revealed differences in visual outcome associated with meningiomas in three anatomical subdivisions (intraorbital, optic canal, and intracranial space). Postoperative visual improvement was 50% with meningiomas of the intracranial segment, 31% with optic canal involvement, and 11% with intraorbital location [64].

For intraorbital lesion outcomes, the authors suggested avoiding surgical resection of optic nerve sheath meningiomas and considering radiotherapy as a lower risk alternative. The authors additionally cautioned that mechanical strains of the orbital canal may inhibit radiotherapy effectiveness; they suggested opening the orbital canal over the entire length of the optic nerve offers the best conditions for visual recovery. Radiation therapy may then be proposed while surveilling remnant evolution [64].

Peters et al. performed an extensive systematic review and meta-analysis in 2023 on both single-fraction SRS and hypofractionated radiosurgery. Most analyzed studies focused on hypofractionated radiosurgery (4/6) and the remaining on single-fraction SRS. In total, 865 patients were analyzed with 427 receiving hypofractionated and 438 single fraction. Tumor control rate was near equivalent between modalities with 95.1% for single fraction and 95.6% for hypofractionated [63]. The most used hypofractionation regiment within the review was 25 Gy in five fractions. Visual stability was slightly better in the hypofractionated group (90.4% single fraction, 95.6% hypofractionated). The single fraction showed better visual improvement (29.4% vs. 22.7%), but also higher rates of visual decline (9.6% vs. 5.1%). The authors concluded SRS was effective and safe, with both hypofractionated and single-fraction SRS considered viable options.

One of the largest studies conducted was a retrospective analysis of 438 perioptic meningioma patients from 12 institutions, carried out by Bunevicius et al. Most patients (92.5%) underwent single-fraction SRS with a median marginal dose of 12 Gy. The median maximal dose to the optic apparatus was 8.5 Gy. Their series found that single-session SRS demonstrated superior tumor control when compared to 2–5 fractions, although the fraction group was small, limiting generalizability of the findings. The study showed that 5 and 10-year progression-free survival was 96% and 89%, respectively [66]. A dose of ≥10 Gy to the optic apparatus and tumor progression were independent predictors of post-SRS visual decline.

Asuzu et al. performed a large multicenter retrospective analysis, of 328 patients at 11 institutions with meningiomas in direct contact with the optic apparatus. More specifically, 107 patients had tuberculum meningiomas, 126 had clinoidal meningiomas, and 105 had cavernous sinus meningiomas. The majority (64.6%) of patients underwent SRS as the initial treatment. Results showed a tumor control rate of 91%, and no difference in visual outcomes between patients whose tumors contacted the optic nerves compared with chiasm or anterior optic tract contact [67]. The authors also found symptom duration predicted SRS response failure irrespective of lesion size. The authors concluded earlier intervention could improve tumor response and that pre-SRS decompressive surgery should be considered for patients with symptomatic compressive optic neuropathy. Within their study, 93% of patients had single-session SRS, the rest fractionated therapy. Their results found no association between fractionation, visual decline, nor tumor response [67].

Chen et al. retrospectively analyzed 60 patients with perioptic tumors from 2007 to 2020. Of the 60 tumors, 53 were meningiomas and 7 schwannomas. Patients received hypofractionated radiosurgery with 6–7 Gy three times daily for three consecutive days (mean optic apparatus dose 6.05 Gy). Observed tumor control rates at 1, 3, 5, 8, and 13-year follow-up were 98.3%, 93.4%, 90.6%, 88.4%, and 88.4% [68]. Patients with pre-existing optic nerve compression, concurrent chemotherapy, and prior radiotherapy faced higher risk of radiation-induced optic neuropathy. The authors concluded that tumor growth within 6 months requires further close monitoring, and if enlargement continues >2 years, further intervention may be required. Overall hypofractionated dosage over three days provided adequate control without significant side effects.

Wei et al. performed a retrospective analysis of seven patients with recurrent optic nerve sheath meningiomas following gross surgical resection. The median maximal radiation to the optic nerve was 6.5 Gy (1.9–8.1 Gy). Their results showed that salvage SRS resulted in tumor control and visual preservation in 5/7 patients [69].

Su et al. performed volume-staged GKS using 2–3 fractions. In the first stage, a higher median marginal dose of 13.5 Gy was applied to the basal portion of the tumor. The second stage involved a smaller 9 Gy isodose to the superior portion proximal to the optic apparatus, or “snowman-shape” volumetric treatment. All four patients demonstrated a 34–46% reduction in tumor volume and improvement of neurological defects to various degrees [70].

El-Shehaby et al. performed a retrospective analysis of 175 patients with large (≥10 cm^3^) perioptic meningiomas. Patients received single-session SRS with a median prescription dose of 12 Gy. The tumor control rate was 92%, and 97% had better or stable visual outcomes [71].

When treating primary optic nerve sheath meningiomas, radiation has achieved superior results in stabilizing or improving vision compared with surgery or observation [72]. An additional technology to treat primary optic nerve sheath meningiomas is intensity modulated radiotherapy (IMRT, an advanced form of three-dimensional conformal radiation therapy investigated for meningioma treatment since the mid-1990s [73]). IMRT allows for optimized dose distribution with sparring of surrounding normal tissues. Prospective studies comparing IMRT with SRS/hypofractionated SRT are warranted and IMRT is not widely available at this time [72].

Reviewed studies suggest both fractionated and single-session radiosurgical intervention offer high control and acceptable complication rates for perioptic meningiomas. This holds true for both upfront and salvage SRS for post-surgical progression. As symptom duration predicted SRS response failure regardless of tumor size [67], earlier intervention may benefit. Fractionated therapy has a smaller risk of radiation-induced optic neuropathy (0–6.7%) versus single-session therapy (13%) [74]. Determining optimal treatment timing and fractionation dosage warrant additional studies.

### 5.6. Petroclival

The petroclival meningioma (PCM) accounts for 11.42% of posterior fossa meningiomas and is located in a particularly difficult area [9]. PCMs typically arise from the upper two-thirds of the clivus, directly adjacent to the brainstem, basilar artery, perforating arteries, and multiple cranial nerves [75]. Pure petroclival meningiomas are not considered to belong to the parasellar region, although cavernous sinus meningiomas commonly extend to the parasellar region [18]. Extending cavernous sinus meningiomas involving the petroclival region should be named caverno-petroclival or spheno-cavernous-petroclival [18]. This section addresses pure petroclival meningiomas.

With adjacent vital neurological and vascular structures, PCMs are one of the most challenging skull base tumors [19]. Microsurgical resection has morbidity ranging from 28% to 76%, most with incomplete resection [21]. Due to a lack of published literature, the use of SRS as treatment for petroclival meningiomas remains relatively undetermined [19]. Similar to other difficult anatomical locations, the optimal management of asymptomatic PCMs remains controversial [20].

Wijaya et al. in 2022 conducted a systematic review of petroclival meningioma patients undergoing SRS [19]. The first to determine the safety and efficacy of SRS treatment of PCMs, their study consisted of synthesized evidence from 10 studies and 719 patients. Their results showed overall a tumor control rate mean of 98.8% (85–100%) and a low number of post-SRS complications [19]. Still, significant selection bias likely exists between primary and adjuvant therapy, impacting the interpretation of results.

Alamer et al. systematically reviewed seven articles involving 722 cases of petroclival meningiomas. Their review found that a mean marginal tumor dose of 13.5 Gy resulted in 5 and 10-year progression-free survival of 91–100% and 69.6–89.9%, respectively [21]. The authors compared outcomes between primary (61.9%) and adjuvant radiosurgery (38.1%) and found that primary SRS reported higher rates of tumor control (94.3% vs. 88.2%) and fewer SRS-related complications (3.7% vs. 10.3%) [21]. The authors concluded that primary SRS was effective for smaller lesions without symptoms related to mass effect.

Mantziaris et al. performed a retrospective, international, multicenter study on asymptomatic PCMs using data from the IMPASSE study. They found that upfront SRS for asymptomatic, petroclival region meningiomas achieved local tumor control in all 79 patients with <5% experiencing new neurological deficits [20]. The authors concluded that SRS could be considered at the diagnosis of PCMs and recommended if any growth is noted during active surveillance [20].

The overall reviewed studies provide evidence that radiosurgery provides excellent long-term control and minimal complications for surgically challenging petroclival meningiomas. SRS has roles in both upfront primary intervention and adjuvant treatment for petroclival meningiomas. We were unable to find any data available on the efficacy of fractionated dosing for PCMs.

### 5.7. Clinoidal

Clinoidal meningiomas comprise less than 10% of supratentorial meningiomas [76] and represent a challenge for neurosurgeons. The clinoidal area has high neurosurgical complexity, with significant resection-associated morbidity and mortality [34]. Group I clinoidal meningiomas grow from the inferior portion of the clinoidal process with frequent ICA attachment, making resection exceedingly difficult. Group II arises from the superior and lateral face of the anterior clinoid process; preservation of the arachnoid layer and the carotid cistern facilitate tumor dissection. Group III arises from the optic canal from the medial side of the anterior clinoid process [18,34]. These tumors often adhere to vessels like the middle cerebral artery, anterior cerebral artery, and internal carotid artery [34,76]. Rates of subtotal resection and subsequent progression are high (41% and 38%, respectively), making SRS valuable [76].

Prior to work by Bunevicius et al., most published studies did not consider the outcomes of clinoidal meningiomas [76]. Their multi-institutional study included 270 patients treated with SRS for clinoidal meningiomas. Most patients underwent single-session treatment, with only 15 having hypofractionated sessions (2–5). Hypofractionated SRS is considered when growths directly abut optic structures and/or large tumors [76]. The median prescription dose was 12 Gy with a maximum of 8.5 Gy to the optic apparatus. Results showed overall tumor control in 93% of patients without significant difference between those treated with single fraction vs. 2–5 fractions of radiation [76]. Visual decline was associated with a dosage of ≥10 Gy to the optic apparatus and the presence of visual impairment prior to SRS.

Two studies investigated anterior clinoid process (ACP) meningiomas specifically [77,78]. Prior to these studies, ACP meningiomas were pooled as sphenoid meningiomas in lieu of individual analysis.

In the first study investigating fractionated radiosurgery on ACP meningiomas, Demiral et al. published on 19 patients who underwent hypofractionated radiotherapy for ACP meningiomas. Volumetric modulated arc therapy was performed with a 6 MV linear accelerator. All patients received 25 Gy total dosage in five fractions over 5 days. Results showed that local control at 1 and 3 years was 100% and 89.4% [77].

In the first study to investigate single-fraction radiosurgery on ACP meningiomas, Akyoldas et al. retrospectively analyzed 61 patients receiving GKRS from 2008 to 2016. With a median marginal dose of 12 Gy, the majority received radiosurgical intervention as primary treatment (80%). Their results showed 100% tumor control, with 61% having tumor regression [78]. Visual impairment was the only preoperative deficit in this cohort; 55% of those patients improved.

Prior to the noted studies, few studies examined clinoidal meningiomas directly. Collectively, these studies demonstrated that radiation for clinoidal meningiomas provides excellent tumor control with acceptable complication rates. Both single-fraction and fractionated intervention were effective on clinoidal meningiomas.

### 5.8. Intraventricular

Intraventricular meningiomas (IVMs) are rare (0.3–5% of meningiomas) [35]. The ventricular system, a series of cerebrospinal fluid (CSF)-filled cavities deep within the brain, cushions surrounding gray and white matter. Due to their deep locations, IVMs are among the most challenging tumors to surgically manage, with significant treatment-associated morbidity [35,79]. The surgical approach requires traversing the cerebral cortex, avoiding critical nerve tracts adjacent to the tumor and along the trajectory, and controlling homeostasis deep inside the brain [35]. IVMs may be closely associated with vital structures like the thalamus, hypothalamus, basal ganglia, internal capsule, and brainstem [80]. They preferentially arise in the lateral ventricle, followed by the third and fourth ventricles [79]. Due to their rarity, few studies describe SRS to treat IVMs.

Kim et al. published the first study of SRS for IVMs in 2009 with a case series of nine patients. Using a median marginal dose of 16 Gy (range 14.0–22.5), they achieved tumor control in seven out of nine patients. The authors concluded that SRS should be considered for small intraventricular tumors with minimal symptoms and as an alternative to repeat surgery for residual or remnant tumors [81].

In 2022, Christ et al. published a pooled analysis of all prior studies with their case series of 33 patients undergoing primary SRS for IVMs. Prior studies had between 2 and 19 patients comparatively [79,80,81,82,83]. With a mean marginal dose of 13.9 Gy to treat mostly lateral ventricular IVMs (97%), results covered a long-term median follow-up of 7.9 years. In total, 66% of patients had a partial or minor response, and the rest had stable disease [84]. Moreover, 91% of patients in a pooled analysis had local control.

Daza-Ovalle et al. updated the field on IVM SRS results in their 2022 case series and literature review. In their case series, 18 of 19 patients achieved symptom control after single-session GKRS, with progression-free survival 95% at five years and 85% at ten years [79], comparable to a Simpson Grade I resection. The authors concluded that early SRS intervention should be considered for the primary management of small to medium IVMs and adjuvant for residual IVMs.

In 2023, Umekawa et al. analyzed thirty years of institutional data in a retrospective observational study that included 11 patients with 12 IVMs treated with SRS. A median marginal dose of 16 Gy was given (range 9–18). Results demonstrated tumor control of 100% (follow-up period 52 months) with 55% decreasing in size [35]. Notably, their cohort included patients with NF2-related IVMs and demonstrated efficacy regardless of NF2 mutation status.

There were three additional smaller studies carried out by Nundkumar (N = 2), Mindermann (N = 5), and Samanci (N = 6) between 2013 and 2020. The mean marginal dose ranged from 12 to 16 Gy, and tumor control was 100% for all of the studies [80,82,83].

Numerous studies were available for IVMs; however, due to their rarity, all were small in patient number. Overall, the studies demonstrated excellent tumor control with an acceptable complication rate for IVMs. Compared to other anatomical sites, IVMs had fewer total cases available. More research in larger studies is warranted. No studies mentioned the use of fractionated radiation for IVM treatment.

### 5.9. Olfactory Groove

Olfactory groove meningiomas comprise 10% of intracranial meningiomas; collectively, they describe tumors arising from midline anterior cranial fossa structures such as the crista galli, cribriform plate, olfactory groove, and planum sphenoidale [24]. Anosmia after olfactory groove meningioma resection can cause significant disability. Rates of olfactory preservation are between 50% and 95% with surgical resection [85]. Radiosurgical intervention is a less invasive option that may promote olfactory preservation.

A study by Gande et al. in 2014 was the first to investigate subjective olfactory function after Gamma Knife SRS. They performed a retrospective analysis of 41 patients who underwent SRS for anterior fossa cranial tumors with more than 6 months of follow-up available. Radiosurgery was performed as primary management for 22 patients and as adjuvant therapy for the remaining 19 patients. Two patients underwent prior fractionated external beam radiation therapy. The mean marginal dose was 13 Gy, and the cumulative progression-free tumor control rates were 97% at 1 year and 95% at 2, 10, and 20 years [24]. Post-SRS, 27 (66%) patients reported intact olfaction, nine (22%) reported partial deficits, and five (12%) patients had no improvement from complete anosmia [24].

A large multi-institutional study was conducted by Bunevicius et al. in 2021 using pooled results from 20 institutions participating in the Radiosurgery Research Foundation. They found 278 patients with more than 6 months of follow-up who underwent SRS with a median dose of 12 Gy. Forty-three percent of patients had partial or marginal response, 54% had stable disease, and 3% experienced progression [22]. After the SRS, olfaction remained stable, improved, or deteriorated in 90%, 8%, and 2% of patients, respectively [22].

Another single-center retrospective cohort study was performed by Liu et al. They reviewed 13 patients treated with single-session SRS (N = 5), hypofractionated stereotactic radiotherapy (HSRT) (N = 6), and fractionated stereotactic radiotherapy (FSRT) (N = 2). SRS was one dose of 10 Gy, HSRT was 25 Gy in five fractions, and FSRT was 54 Gy in 30 fractions. The median prescribed dose was 14.8 Gy for SRS, 27.3 Gy for HSRT, and 50.2 Gy for FSRT. The overall median maximal dose was 32.27 Gy. They achieved 100% regional control rate in 12 out of 13 patients, and 6 out of 12 had reduced tumor volume [86]. Every patient included in the study had stable vision and olfactory function at the time of the last follow-up (median follow-up 48 months). The authors highlighted the advantages of using fractionated regimens for cranial nerve sparing when tumors are spatially associated with the optic apparatus or >10 cm^3^. Zaorsky et al. reviewed 13 patients treated with 52.8 Gy over 25 fractions or 16.1 Gy in a single fraction. The authors concluded that both FSRT and SRS have a higher likelihood of preserving or improving cranial nerve function and decreasing morbidity compared with operative intervention [87].

Radiosurgical intervention for olfactory groove meningiomas showed comparable preservation of olfaction with surgical intervention and acceptable control rates among all studies reviewed. Fractionated and unfractionated radiation are both viable options that may offer better preservation or improvement of cranial nerve function.

### 5.10. Cerebellopontine

Cerebellopontine angle (CPA) meningiomas comprise 1% of all intracranial meningiomas [88], but they are the second most common tumor located at the CPA after vestibular schwannomas [36]. These tumors arise from the petrous face of the temporal bone and are categorized into anterior, middle, or posterior based on their relationship with the internal auditory meatus [89]. This area is compactly filled with neurovascular elements, including cranial nerves IV to XII, vertebral, and basilar arteries [90]. Meningiomas are the minority of tumors at the CPA, with vestibular schwannomas being the most common tumor in this area. When compared directly, it was observed that hearing preservation was better for CPA meningiomas when compared to vestibular schwannomas [91]. There has been research showing increased post-surgical facial paralysis in tumors located anterior (60%) or below (50%) the internal auditory canal (IAC) compared with those located posterior or superior (15%). Considering the anatomical risk, radiosurgical intervention is a viable option to avoid surgical morbidity.

Gendreau et al. performed a systematic review and meta-analysis analyzing surgical outcomes and radiation dosages. Their review found six studies, including 406 patients who underwent radiosurgery for CPA meningiomas. With a median marginal dose of 12–15 Gy, they found minimal cranial nerve complications while having an overall tumor control rate of 95.6% [36]. Postprocedural tumor regression was associated with a median prescription dose of >13 Gy.

Another retrospective descriptive analysis was performed by Garcia et al. on 80 patients with CPA meningiomas from 2001 to 2014. SRS was the primary treatment in the majority of the cases (83.7%). All patients received a single-dose treatment, and the median coverage dose was 14 Gy. After an extensive 12-year follow-up, tumor control rates were found to be 95% [92]. Radiation dose to the brainstem was related to the symptomatic deterioration in four patients.

Park et al. investigated 74 patients with CPA meningiomas from 1990 to 2010. Patients underwent single-session SRS with a median prescription dose of 13 Gy. Tumor extension into the IAC was related to preexisting hearing loss (*p* = 0.01). The median radiation dose to the cochlea was 3.2 Gy in patients with unchanged hearing and 2.5 Gy in patients with improved hearing. Control rates were found to be 97% [93], although the average follow-up time of 40 months was comparatively shorter than Garcia’s study. Their results additionally showed that patients with trigeminal neuralgia related to CPA tumors are more likely to experience pain worsening over long-term follow-up.

Another large retrospective review was performed by Jahanbakhshi et al. on 93 CPA meningioma patients. Patients received a median marginal dose of 13.6 Gy. Progression-free survival was seen in 96% of patients [90]. Their median tumor volume was 6 cm^3^, twice as large as Park’s 3 cm^3^ median tumor volume and Garcia’s average tumor volume of 3.12 cm^3^. Their results demonstrated adequate tumor control rates and minimal adverse radiation events even with tumors of larger size. They additionally found that male sex was associated with lower progression-free survival and tumor volume ≥ 8.5 cc was associated with worse symptomatic outcomes.

Overall, there was a larger number of patients to draw conclusions from for CPA meningiomas compared with other anatomical regions. The studies reviewed indicate that radiation therapy is effective at long-term control with acceptable complication rates. Tumors located anterior or below the IAC, with a higher risk of facial paralysis, may benefit from radiotherapy. A median dose of >13 Gy is recommended for tumor regression, and radiation to the brainstem should be avoided at all costs.

### 5.11. Foramen Magnum

Foramen magnum meningiomas (FMMs) represent only 2–3% of intracranial meningiomas; hence, treatment outcomes and optimal management paradigms remain poorly defined [28]. Surgical intervention is extremely challenging for neurosurgeons due to their intimate proximity to critical structures, including the brainstem, lower cranial nerves, medulla, vertebral artery, and its branches [26]. Anterior and anterolateral FMMs carry the highest surgical risk and are the most common location for FMMs [28]. Considering the highly eloquent location of FMMs, radiation may be a viable alternative for primary management in addition to handling residual/recurrent tumors.

Karras et al. performed the first systematic review of the literature on FMMs in 2022, contributing their own institutional knowledge alongside. Their review found nine patients from the authors’ institution and 165 patients accrued from four other prior publications. SRS was utilized as the primary therapy in 63.6% of patients and as salvage (21.8%) or adjuvant (14.5%) therapy for the remaining patients [25]. Clinical stability and local control at the last follow-up were achieved in 98.8% and 97.0% of patients, respectively [25]. The researchers concluded that SRS may be a viable primary option for symptomatic lesions without significant brainstem compression and for higher-risk surgical patients. They stated that larger tumors (>35 mm in diameter) should undergo maximal safe resection, and SRS is strongly considered in cases of subtotal resection [25].

Akyoldas et al. performed the largest from a single institution and longest follow-up period to date on FMMs in 2021. Researchers performed a retrospective analysis of 37 FMM patients undergoing Gamma Knife radiosurgery, 12 of which had prior microsurgical resection. Their results showed GKRS provided excellent tumor control with a rate of 97.3% during a median radiological follow-up of 7 years [26]. Results from their cohort demonstrated that GKRS is an effective and safe treatment for patients with both primary and recurrent/residual FMMs [26].

Recent publications suggest that despite historic underutilization of SRS for FMMs, it is recently gaining traction and recognition as an important treatment option [25]. Both the systematic review by Karras and institutional analysis with long-term follow-up by Akyoldas suggest radiation is a viable option for long-term control.

## 6. Discussion

The gold standard for meningioma treatment remains complete surgical removal of the tumor and adjacent dura/bone involved. Advances in neurosurgical technique and technology permit better gross total resections than ever. However, as radiation therapy technology has advanced, one has to balance the ability to achieve a total resection against the associated morbidity of the particular procedure. Alongside advances in operative technique, our understanding of radiation efficacy and application builds with additional publications. For meningiomas in both high and low-risk locations, the role radiation plays in management is continually evolving. It is increasingly clear that a “hybrid” approach using resection followed by RT can significantly reduce morbidity in certain high-risk situations. Adaptive hybrid surgery analysis (AHSA) is an intraoperative tool used for the automatic assessment of tumor properties. The software provides an immediate assessment of adjuvant radiation coverage options for single-fraction and hypofractionated radiation based on an IMRT algorithm [94].

### 6.1. Observation, Radiation, or Surgery

The course of treatment remains dependent on individual patient characteristics, including age, life expectancy, location-specific risk, and tumor characteristics. For lesions that are symptomatic, Grade II or III, or in dangerous locations, observation is an unreasonable strategy, and intervention of some form is recommended [6]. Gross total resection remains the gold standard for symptomatic accessible lesions in patients who can tolerate surgery. When discussing management for small asymptomatic lesions, there is less clarity on the optimal management. A meta-analysis of 27 studies was performed to formulate practice guidelines for non-cavernous sinus benign intracranial meningiomas in 2020 [95]. The analysis showed that although the current literature lacks level I and II evidence supporting SRS for benign intracranial meningiomas, the large quantity of level III studies justifies recommending SRS as an effective primary treatment option for Grade I meningiomas (recommendation level II) [95].

Incidentally discovered meningiomas now account for 30% of newly diagnosed intracranial meningiomas [96]. Conventional wisdom has favored observation over radiosurgical intervention with small and asymptomatic meningiomas. This approach is favorable for elderly patients, those with significant comorbidities, and poor performance status [65]. A recent meta-analysis performed on 2130 patients with incidentally discovered meningiomas showed that the initial management strategy at diagnosis was active monitoring (50.7%), surgery (27.3%), and stereotactic radiosurgery (22.0%) [96].

The watch-and-scan paradigm is being challenged as further studies suggest better tumor control and fewer side effects associated with radiosurgical intervention [97]. Upfront SRS may be a better option for younger patients where tumor progression will inevitably require active treatment [65]. Additionally, some studies have suggested that earlier intervention, at younger ages, is favorable for progression-free survival [10]. As further research continues to amass favoring SRS for long-term control, we may see guidelines change in the future of meningioma management.

Although younger patients are good candidates for upfront SRS intervention, certain patient populations may be at higher risks of developing iatrogenic complications. One of the paradoxical risks of radiation treatment of meningiomas is radiation-induced meningiomas growing as a result of treatment. There are limited data on the treatment of radiation-induced meningiomas, but they are more often atypical and highly proliferative [65].

Recent studies have uncovered gene mutations that may contribute to the development of radiation-induced meningiomas. The presence of a mutation in Cycline D1, p16, 1p, and 22q were more commonly associated with radiation-induced meningiomas [6]. As we further our understanding of pre-existing risk factors for radiation-induced meningioma development, it may be used to suggest surgical intervention or observation over radiation. This is especially true in younger patients or those who are likely to have multiple tumors (NF2 patients).

The anatomical challenges are mainly based on the proximity of the tumor to critical structures such as the optic apparatus or brainstem. When the radiation dose constraints are exceeded for one or more of the critical structures, the utilization of a more protracted radiosurgical regimen (HSRT) allows for delivery of a radiobiologically equivalent dose to the tumor when respecting the constraints of the critical structures. Therefore, HSRT is frequently used to treat meningiomas in eloquent locations.

Advances in predicting an individual tumor’s growth trajectory would aid in choosing between SRS and observation. Currently, tumor size ≥ 3 cm, peritumoral edema, young age (<60 years old), lack of calcifications, and lesion hyperintensity on T2-weighted MRI were found to be significantly related to symptomatic progression [65]. A risk calculator developed to determine progression risk based on tumor location, molecular markers, radiogenomics, and individual patient comorbidities would be of high clinical utility. There are studies investigating MRI-based machine-learning tools for diagnosis, prognostication, and automatic segmentation of meningiomas; these tools are in the early stage of research but may serve as radiomic-based predictive models of aggressiveness in the future [98].

### 6.2. Radiation Dose and Technique

Historically, stereotactic radiosurgery was exclusively delivered using a frame-based system using Leksell Gamma Knife which can create a very steep dose gradient beyond the treated target [99]. However, the older versions were limited by the need to set up the patient manually and forward treatment planning. Furthermore, the target in peripheral and inferior locations might not be readily reached due to the physical limitations of the machine. Furthermore, the frame-based nature of Gamma Knife limited the dose delivery to one fraction which could limit its ability to treat meningiomas adjacent to critical structures. Subsequently, LINAC-based systems (adapted linear accelerators) have been developed, allowing for hypofractionated stereotactic treatments to be given with the use of a relocatable headframe and there was no limitation in the target location [100]. The development of a robotic radiosurgery system (CyberKnife) allows for near real-time tracking of the skull using a mask-based system using inverse planning [101]. Over the years, tremendous improvements have been made on the Gamma Knife system which is now capable of delivering hypofractionated treatments using masked-based immobilization, cone-beam CT, and infrared tracking [102]. Furthermore, selective shaping by blocking one or more of the sectors is possible. The experience of the treatment team is likely more important than the treatment device itself provided that the latter possesses the basic features allowing for the delivery of a radiosurgical dose with a rapid fall-off of the dose beyond the target.

Radiation dosages should be optimized to reduce postprocedural morbidity while also achieving high rates of tumor control [36]. Commonly used dose regimens for WHO Grade I, II, and III meningiomas treated with single-fraction SRS are 12–16 Gy, 16–20 Gy, and 18–24 Gy, respectively [63]. Although, in patients with recurrent WHO Grade II meningiomas the optimal modality of radiation is not well-established [103]. The most common radiation doses, techniques, and planning used in studies reviewed within prior sections are reported in Table 2 for comparison. In general, SRS and HSRT are not regarded as an appropriate treatment for WHO Grade III meningiomas; larger resection margins are typically needed to avoid recurrence. Both WHO Grade II and III lesions typically require a combination of maximum surgical resection and radiation [104]. The European Association of Neuro-Oncology (EANO) issued the first guidelines for diagnosis and treatment for meningiomas in 2016; this was subsequently updated in 2021 [16]. Updated guidelines rely on a combination of histopathology and molecular pathology, which is typically not available until weeks after surgical resection and tissue sampling. Once the final pathology is available, the WHO classification can be formally attached to the diagnosis and aid in treatment decisions. Choosing to go forward with radiation after resection typically occurs in an outpatient follow-up visit for most patients.

Radiosensitive structures such as the optic apparatus, brainstem, cochlear nerve, and other cranial nerves should be taken into consideration. Damage to the optic apparatus may cause radiation-induced optic neuropathy, which may manifest months or years later with painless visual loss, changes in color vision, and pupillary abnormalities [63]. The risk of developing clinically significant radiation optic neuropathy was found to be 1.1% for those who received ≤12 Gy [105]. Using fractionated therapy doses as high as 54 Gy in total, delivered in 6–30 fractions with 1.8–4 Gy per fraction, has demonstrated minimal visual deterioration (1.5% of patients) [74].

Fractionating the radiation intends to decrease the risk of radiation damage to normal tissues and allow for increased dose delivery to larger tumor volumes. The fractionation decreases the risk of complications but does not decrease the overall radiation exposure. Conventionally fractionated radiation takes advantage of differing cell repair potentials between healthy tissue and tumor, allowing the healthy tissue to repair between spread-out sessions [106]. Hypofractionated and single-session SRS require increased conformity and dose-gradient steepness to avoid damage to radiosensitive tissues [106]. Typically for Grade I meningioma, the recommended fractionated dose is 50.4–54 Gy in 1.8 Gy per fraction.

For region-specific treatments discussed within this review, there was more evidence available for single-fraction radiosurgery than HSRT. Current guidelines support the use of conventionally fractionated radiotherapy over SRS for WHO Grade II and III meningiomas; for WHO Grade I, either SRS or conventionally fractionated radiotherapy is a suggested option [16]. Other guidelines suggest fractionated radiation is beneficial in tumors > 7.5 cc and <3–5 mm from radiosensitive structures [106].

A systematic review and meta-analysis including 1736 meningioma patients compared single-session SRS, hypofractionated radiation, and conventionally fractionated radiation. Their results showed both were safe options, but fractionation had superior radiographic control and a lower incidence of symptomatic decline and edema [107].

Another publication compared conventionally fractionated with hypofractionated radiation among 341 patients with skull base meningiomas. Pooled multicenter analysis demonstrated no difference in control rates between fractionation patterns and comparable rates to SRS [106].

**Table 2 cancers-17-00045-t002:** This table shows the radiation parameters reportedly used among anatomical regions. Any region with reported fractionated dosing was also included and denoted N/A.

Anatomical Location	Mean Marginal Dose (Single Session)	Mean Marginal Dose (Fractionated)
Parafalcine/Parasagittal	14–18 Gy [45]	N/A
Convexity	13.26 Gy [48]	25 Gy (5 fractions) or 50.4 Gy (28 fractions) [108]
Cavernous Sinus	12–14 Gy [55]	51.2 Gy (15.5 sessions) [109]
Parasellar	12–14 Gy [59,61]	25–30 Gy (5–6 sessions) [33]
Perioptic	10–12 Gy [63]	25 Gy (5 sessions) [63]
Petroclival	15 Gy [19]	N/A
Clinoidal	12 Gy [76,78]	25 Gy (5 sessions) [77]
Intraventricular	12–16 Gy [79]	N/A
Olfactory Groove	12 Gy [22]	25 Gy (5 sessions) [86]
Cerebellopontine Angle	12–15 Gy [36]	N/A
Foramen Magnum	12–14 Gy [25]	20 Gy (5 sessions) [25]

### 6.3. Predicting Response to Radiation

Understanding which meningiomas are responsive to radiation enables informed surgical decision-making. WHO Grade I meningiomas demonstrate a much better response to radiation than Grade II or III meningiomas [110]. The only method currently available for a definitive diagnosis of WHO grade is a tissue biopsy, and the question of which patients warrant an upfront biopsy has yet to be answered [16,54]. Sughrue et al. suggested that patients with immunosuppression, nasopharyngeal or infratemporal fossa extension, rapid symptom onset, cavernous sinus syndrome, or other synchronous lesions are indications for a tissue diagnosis [54].

The standard postoperative treatment for intermediate risk meningiomas (WHO Grade II or recurrent Grade I) is conventionally fractionated radiotherapy irrespective of resection extent [111]. Radiotherapy including the remaining enhancing tumor and an additional volumetric expansion of 10 mm accounts for any microscopic subclinical disease extension [111]. For high-risk meningiomas (Grade III, recurrent Grade II), trials have investigated radiation expansion up to 1–2 cm in margin [112].

Trial NRG-BN003 (NCT 03180268) suggests that completely resected Grade II meningiomas may be watched without adjuvant radiation [113]. In a prospective observational study of patients with Grade II meningiomas who underwent gross total resection, 2/12 patients were found to have progression. Both were successfully salvaged with focal fractionated radiosurgery after recurrence.

Variability of oncologic outcomes exists within each grade; even Grade I meningiomas have certain subsets that will recur [114]. Postoperative radiation practice patterns for subtotally resected WHO Grade I or totally resected Grade II meningiomas also remain varied among providers [115]. Prior research has suggested the use of recursive partitioning analysis (RPA) to select optimal candidates for SRS therapy in patients with Grade II meningiomas [116]. An analysis using RPA of 230 patients with Grade II meningiomas at 12 institutions demonstrated that patients age ≤ 50 years old, with up to one prior resection, and no prior radiation therapy are within a good-prognostic group and are likely appropriate candidates for adjuvant SRS [116].

The WHO grades meningiomas are based on results from tissue sample histology and tumor features. In the 2016 updated WHO classification, tumors with a mitotic count greater than four or brain invasion are classified as WHO Grade II, or an atypical meningioma [117]. The update in 2021 included molecular factors that significantly shorten survival and increase recurrence, such as meningiomas with a telomerase reverse transcriptase (TERT) promoter mutation or homozygous deletions in the cell cycle regulator genes CDKN2A and/or CDKN2B [6]. Presence of either TERT or CDK2A/2B will earn a Grade III classification. Future research is warranted to develop clinical biomarkers for both progression and prognostication of meningioma behavior.

### 6.4. Timing of Radiation Intervention

Radiation can be delivered as monotherapy, adjuvant therapy following resection, or after recurrence. The timing of SRS may have an impact on tumor control. The large multicenter retrospective analysis conducted by Asuzu et al. suggested that when upfront SRS is used, earlier intervention from diagnosis could improve the odds of tumor response [67]. Another previous study of 238 WHO Grade I meningiomas investigated the timing of post-resection SRS on symptom control. The authors found excellent long-term control and concluded that earlier radiosurgery was associated with superior symptomatic improvement (*p* = 0.007) [118]. Large prospective multicenter studies are warranted to determine the optimal timing for SRS intervention.

### 6.5. Limitations

This narrative review article has several limitations. Narrative reviews, which lack strictly governed search guidelines, can be more prone to bias than a systematic review. To minimize this limitation, we included all MEDLINE search articles that met our inclusion criteria. Still, the most rigorous review of the evidence would be a meta-data analysis. Other limitations include the lack of formal comparative analysis with existing standard treatments, which would be possible in a meta-data analysis. A true comparison of treatment methods would be most rigorous if it were a meta-data analysis comparing randomized clinical trial results. Unfortunately, the rarity of these anatomically unique meningiomas is at the crux of these limitations. Rare tumors mean rare data. Continued efforts at retrospective analysis, and clinical trials where possible, can yield higher level evidence for optimal treatment determinations.

Our review also did not discuss the malignant transformation of meningioma. How is radiosurgery relevant in malignancy? This question warrants additional research, and a review focused on this specific topic, if there is sufficient literature available. This review additionally lacked long-term follow-up results for radiation therapy. Many studies reviewed were within the past 5 years and long-term follow-up outcomes have not been published or are simply not available yet. A follow-up specifying long-term outcomes by a specific anatomical site would be a great opportunity for future publication when available.

Additionally, our review did not individually divide up patient characteristics such as age and tumor size throughout the review. Some articles referenced only the discussed surgical management of meningioma; these were presented to draw comparison with radiosurgical intervention within the same anatomical subregion. Tumor dimensions were mentioned when highlighted as a significant factor in study results. A more detailed analysis of tumor dimension would be an important follow-up review.

### 6.6. Future Treatment Horizons and Technological Advances

Considering that meningiomas reside outside of the BBB, they are accessible to systemic therapies and circulating immune cells [114,119]. Currently, systemic therapy is regarded as having a minimal role in the management of meningiomas. There has been evidence suggesting links between lymphocyte infiltration and mutational burden in meningiomas; authors have suggested a role for adjuvant immunotherapy in subsets of amenable tumors that cannot be grossly resected [114]. There is strong evidence suggesting PD-L1 correlates with WHO grade and possible tumor recurrence [119]. There are ongoing clinical trials investigating PD-L1 checkpoint inhibitors (NCT02648997, NCT03173950, NCT03604978, NCT03279692, NCT03016091, NCT04659811) and PD-L1 inhibitors (NCT03267836). There are also a wide variety of molecular mutations being targeted for systemic therapy [30]. Classic cytotoxic agents, somatostatin analogs, and anti-hormone therapies have shown limited efficacy thus far; tyrosine kinase inhibitors and monoclonal antibodies targeting angiogenesis have shown promise in small phase II trials [120].

## 7. Conclusions

Meningioma management is individualized based on patient comorbidities, tumor location/characteristics, symptomatic burden, and patient age. The gold standard of care remains gross total resection, including bone and dura. SRS publications have demonstrated an expanded utility in meningioma management. In addition to stereotactic radiosurgery’s established role for surgically inaccessible, recurrent, and highly selected Grade II meningiomas, the use of SRS as upfront management for small asymptomatic meningiomas is increasingly investigated. As evidence becomes more regionally specific, we are better enabled to make informed surgical decisions. For all subregions reported, radiosurgical intervention resulted in high tumor control rates and acceptable adverse radiation events. Depending on anatomical location, SRS may be a key tool to complement open surgical intervention or an alternative option as monotherapy.

## Figures and Tables

**Figure 1 cancers-17-00045-f001:**
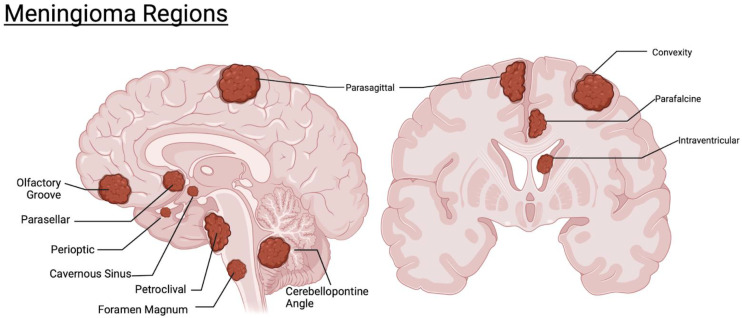
This figure depicts anatomical regions of interest covered within this review. Convexity meningiomas being of the more superficial and surgically accessible compared with cavernous sinus and intraventricular meningiomas.

**Table 1 cancers-17-00045-t001:** This table lists the most common presenting symptoms from intracranial meningiomas arising at various specific anatomical locations. Non-specific symptoms such as headache were excluded.

Table 1.	Symptomatic Lesional Effects
Parasagittal/Falcine	Anterior: Personality change, confusion, and altered consciousness [1], reduced attention span, short-term memory deficits, emotional instability, and apathy [31] Middle: Motor symptoms (commonly spastic weakness of contralateral foot and leg), sensory symptoms, partial seizures [31] Posterior: Visual alterations or headache [31]
Confluence of Falx and Tentorium	Headache, disequilibrium, visual dysfunction, impaired cognition or behavioral change, and upgaze paresis [32]
Convexity	Seizures and sensory changes [12]
Parasellar	Visual impairment [33] (additional symptoms are widely variable depending on subregional location)
Suprasellar	Asymmetric visual impairment [18]
Petroclival	CN V deficits [19], CN VI palsy, vertigo, dizziness, balance and walking disturbances, and weakness of the extremities [18]
Clinoidal	Visual disturbances [34]
Perioptic	Visual loss
Cavernous Sinus	Oculomotor nerve palsy [18]
Intraventricular	Visual field deficits, ataxia, paresis, seizure, and hydrocephalus [35]
Foramen Magnum	Dizziness/ataxia, cranial nerve deficits, numbness, weakness, and hydrocephalus [25]
Olfactory Groove	Mental status disturbances and headaches, partial or complete anosmia [24]
Cerebellopontine Angle	Tinnitus, hearing loss, dizziness, trigeminal dysfunction, neurocognitive impairment [36]

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
