# Peer review of "Radiosurgery for Intracranial Meningiomas: A Review of Anatomical Challenges and an Update on the Evidence"

_cancers, 2024, doi:10.3390/cancers17010045_

Round 1

Reviewer 1 Report

Comments and Suggestions for Authors

 Recommendation: Reject

While the authors commendably attempt to synthesize recent developments in radiosurgery for intracranial meningiomas with a focus on anatomical subregions, the review unfortunately falls short in several critical areas that undermine its utility. First, the methodology for selecting the 'highest levels of evidence' for analysis is not sufficiently transparent or rigorous, which casts doubt on the validity of the conclusions drawn. Additionally, the article lacks a comparative analysis with existing standard treatments, which is essential for contextualizing the efficacy and safety of radiosurgery. The discussion of patient outcomes predominantly emphasizes positive aspects without adequate consideration of long-term follow-up data, which is vital for assessing the true impact of radiosurgery. Moreover, the review could benefit from a more detailed exploration of technological advancements in imaging and radiosurgical equipment, as these are crucial for understanding current capabilities and limitations. Lastly, while the paper aims to provide updates based on specific anatomical challenges, it does not sufficiently address how these challenges impact the practical application of radiosurgery across diverse patient demographics. 

Author Response

Comment 1: The methodology for selecting the 'highest levels of evidence' for analysis is not sufficiently transparent or rigorous, which casts doubt on the validity of the conclusions drawn. Additionally, the article lacks a comparative analysis with existing standard treatments, which is essential for contextualizing the efficacy and safety of radiosurgery.

Response:  We thank the reviewer for highlighting an important limitation of our narrative review. To address this concern, we have modified our claim on the level of evidence and clarified our methodology by adding detailed description of the review style in our methods section (lines 29-31, 404-411). Additionally, we have added a discussion of the limitations of narrative reviews as a methodology (lines 2767-2777). We have also added discussion of the lack of statistical comparative analysis, noting that this article is limited by its narrative methodology (as opposed to a systematic review).

Comment 2: The discussion of patient outcomes predominantly emphasizes positive aspects without adequate consideration of long-term follow-up data, which is vital for assessing the true impact of radiosurgery. 

Response 2: We appreciate the reviewer’s important perspective in addressing the long-term impacts of radiosurgery. Assessing long-term follow-up data in a retrospective or prospective trial is certainly needed for a comprehensive assessment of this treatment. We discussed long-term outcomes as they were reported within the individual studies reviewed. Many of the studies referenced were published to recently for long-term outcomes to be available or was not published. The lack of long-term outcome availability has been added within our limitation section and would make for an excellent follow up publication when data becomes available.  

Comment 3 Moreover, the review could benefit from a more detailed exploration of technological advancements in imaging and radiosurgical equipment, as these are crucial for understanding current capabilities and limitations.

Response 3: We thank the reviewer for this suggestion. We have added a paragraph on technological advancements (lines 2661-2679).

Comment 4: Lastly, while the paper aims to provide updates based on specific anatomical challenges, it does not sufficiently address how these challenges impact the practical application of radiosurgery across diverse patient demographics. 

Response 4: The reviewer points out an excellent way to connect the different strains of thought. We have added a discussion on how anatomical challenges influence radiosurgery implementation for a broad patient population in lines 2644-2649. We additionally commented on radiosurgical intervention in specific subpopulations of patients with neurofibromatosis 2 adjacent to this added paragraph.

Reviewer 2 Report

Comments and Suggestions for Authors

This review described radiosurgery for intracranial meningioma and discussed each tumor location. 

Some reviews discussed only surgery, so these reviews should be omitted.

While tumor location is important, malignancy is also a critical factor in determining treatment strategy. The author should discuss malignancy.

Author Response

Comment 1: This review described radiosurgery for intracranial meningioma and discussed each tumor location. Some reviews discussed only surgery, so these reviews should be omitted.

Response 1: We thank the reviewer for noting that some reviews only discussed surgery. These were included to act as a comparison for radiosurgical intervention at these anatomical regions. We have added a discussion on the limitations of those articles, and that their inclusion can only comment on surgery as a reference (lines 2802-2804).

Comment 2: While tumor location is important, malignancy is also a critical factor in determining treatment strategy. The author should discuss malignancy.

Response 2: The reviewer makes an excellent point. To address this point, we added a discussion on malignant transformation of meningioma as a limitation of our review’s scope and a point for future research. (lines 2787-2789).

Comment 3: Malignancy is also a critical factor in determining treatment strategy. The author should discuss malignancy.

Response 3: See response 2 and manuscript changes (lines 2716-18).

Reviewer 3 Report

Comments and Suggestions for Authors

The paper is a good analysis with latter considerations in terms of diagnosis and outcomes.

We would like to suggest to shorten when possible the manuscript and eventual propose an algorithm pathology based.

Author Response

We would like to suggest to shorten when possible the manuscript and eventual propose an algorithm pathology based.

Response: We agree with Reviewer 3; In the changes-tracked manuscript, we have streamlined our writing style. We edited much of the language throughout the article to shorten the length as requested.

Thank you for the suggestion. We agree that including discussion of the EANO guidelines for meningiomas would be an important addition. The EANO guideline updated paper from 2021 was referenced several times throughout the article (lines 655, 677, 2710). We additionally added a short paragraph describing the updated criteria in 2021 including delayed pathology analysis and how this fits within the clinical context for radiation decision making (lines 2678-2685). 

Reviewer 4 Report

Comments and Suggestions for Authors

The Authors present a descriptive review regarding radiosurgery for the management of intracranial meningiomas, with focus on the features related to different anatomical regions. The manuscript is well-written and the overall quality of the references is good, since updated data are presented. The role of radiosurgery in the management of intracranial meningiomas, especially grade I lesion is still widely debated and this paper gives general informations regarding the actual state of the art. However there are some issues to be solved:

- There are no data regarding how the Literature research was conducted: as stated before, the Authors prepared an extensive analysis of the Literature regarding this topic, but without specifying how the review was prepared. A matherial and methods section is completely missing.

- The presented data highlights the importance of grading, clinical presentation and localization, however there is a major factor that is left out from the analysis (or in some cases described briefly): dimension of the lesion. It's true that the Authors state often that decision of performing radiosurgery must be taken case by case and that dimension is obviously a reason. To increase the quality of the informations given in this paper more data regarding tumor's dimension must be included. 

- Age is another factor that should be considered in deciding the ideal management of intracranial meningiomas. The Authors should take in account this factor as well. 

Author Response

The Authors present a descriptive review regarding radiosurgery for the management of intracranial meningiomas, with focus on the features related to different anatomical regions. The manuscript is well-written and the overall quality of the references is good, since updated data are presented. The role of radiosurgery in the management of intracranial meningiomas, especially grade I lesion is still widely debated and this paper gives general informations regarding the actual state of the art. However there are some issues to be solved:

Comment 1: There are no data regarding how the Literature research was conducted: as stated before, the Authors prepared an extensive analysis of the Literature regarding this topic, but without specifying how the review was prepared. A material and methods section is completely missing.

Response 1: We thank the reviewer for their kind feedback. We also appreciate their astute suggestion, in concert with the other reviewers, to address the methodology used in writing this narrative review. We have added a Methods section (lines 414-420).

Comment 2: The presented data highlights the importance of grading, clinical presentation and localization, however there is a major factor that is left out from the analysis (or in some cases described briefly): dimension of the lesion. It's true that the Authors state often that decision of performing radiosurgery must be taken case by case and that dimension is obviously a reason. To increase the quality of the informations given in this paper more data regarding tumor's dimension must be included. 

Response 2: This is an excellent point. When published found size and dimension to be significant in their results and discussion, they were highlighted within our review (Line 1700, 1736, 2541, 2647). In cases were the reviewed article did not provide dimension as an impact factor on their results, we did not provide details. This was added to our limitations section (lines 2804-2806).  

Comment 3: Age is another factor that should be considered in deciding the ideal management of intracranial meningiomas. The Authors should take in account this factor as well. 

Response 3: Again, this Reviewer brings up a crucial factor - age - that could impact meningioma management. We have added age discussion when it was a significant component of studies reviewed (lines 1253, 2615, 2936), and have also discussed studies not including age as a limitation (lines 2801).

Round 2

Reviewer 1 Report

Comments and Suggestions for Authors

Accept in present form

Author Response

We appreciate you accepting the paper in its present form. Thank you again for your time and diligence in correcting deficiencies within the paper. 

Reviewer 2 Report

Comments and Suggestions for Authors

I recognized significant improvements of this manuscript. 

Author Response

We appreciate you recognition of the improvements we made. We hope the revisions have increased the clarity of the research presented. Thank you again for your time and advice.